# Differential Potential of *Phytophthora capsici* Resistance Mechanisms to the Fungicide Metalaxyl in Peppers

**DOI:** 10.3390/microorganisms8020278

**Published:** 2020-02-18

**Authors:** Weiyan Wang, Xiao Liu, Tao Han, Kunyuan Li, Yang Qu, Zhimou Gao

**Affiliations:** 1College of Plant Protection, Anhui Agricultural University, 130 West of Changjiang Road, Hefei 230036, China; wangwy1614@163.com (W.W.); liuxiaolpl@163.com (X.L.); likunyuan9398@163.com (K.L.); 18738389820@163.com (Y.Q.); 2School of Life Sciences, Anhui Agricultural University, 130 West of Changjiang Road, Hefei 230036, China; 3School of Horticulture Landscape Architecture, Henan Institute of Science and Technology, East Section of Hualan Avenue, Xinxiang 453003, China; hantao402131837@126.com

**Keywords:** pepper, *Phytophthora capsici*, mutant strain, metalaxyl resistance, comparative transcriptome

## Abstract

Metalaxyl is one of the main fungicides used to control pepper blight caused by *Phytophthora capsici*. Metalaxyl resistance of *P. capsici*, caused by the long-term intense use of this fungicide, has become one of the most serious challenges facing pest management. To reveal the potential resistance mechanism of *P. capsici* to fungicide metalaxyl, a metalaxyl-resistant mutant strain SD1-9 was obtained under laboratory conditions. The pathogenicity test showed that mutant strain SD1-9 had different pathogenicity to different host plants with or without the treatment of metalaxyl compared with that of the wild type SD1. Comparative transcriptome sequencing of mutant strain SD1-9 and wild type SD1 led to the identification of 3845 differentially expressed genes, among them, 517 genes were upregulated, while 3328 genes were down-regulated in SD1-9 compared to that in the SD1. The expression levels of 10 genes were further verified by real-time RT-PCR. KEGG analysis showed that the differentially expressed genes were enriched in the peroxisome, endocytosis, alanine and tyrosine metabolism. The expression of the candidate gene XLOC_020226 during 10 life history stages was further studied, the results showed that expression level reached a maximum at the zoospores stage and basically showed a gradually increasing trend with increasing infection time in pepper leaves in SD1-9 strain, while its expression gradually increased in the SD1 strain throughout the 10 stages, indicated that XLOC_020226 may be related to the growth and pathogenicity of *P. capsici*. In summary, transcriptome analysis of plant pathogen *P. capsici* strains with different metalaxyl resistance not only provided database of the genes involved in the metalaxyl resistance of *P. capsici*, but also allowed us to gain novel insights into the potential resistance mechanism of *P. capsici* to metalaxyl in peppers.

## 1. Introduction

*Phytophthora capsici*, an important plant-pathogenic oomycete of *Phytophthora*, damages various crops including members of Solanaceae (such as pepper and tomato) and Cucurbitaceae (such as cucumber and pumpkin) [1]. Pepper blight caused by this pathogen was first discovered in the United States in 1918, and the disease can be catastrophic in a short time, endangering the entire growth period of pepper [2]. At present, metalaxyl is the main fungicide used for preventing and controlling plant diseases such as pepper blight. However, because this type of agent is a site-specific inhibitor and has a single site of action on pathogens, *Phytophthora* can develop resistance as they are susceptible to mutations [3,4,5]. Metalaxyl resistance is mainly found in *Phytophthora* species such as *P. capsici*, *P. infestans*, and *P. parasitica*. A metalaxyl-resistant strain of *P. infestans* was discovered in the Netherlands in 1980 [6]. Subsequently, European and American countries reported metalaxyl-resistant strains of *P. capsici* [7,8,9]. China also reported resistant *P. capsici* in the 1960s [10]. To date, metalaxyl-resistant strains of *P. capsici* have been successively found in Anhui, Gansu, and Yunnan Provinces [10,11,12,13].

There are some studies on the resistance mechanism of *Phytophthora* to metalaxyl. Chen et al. [14] revealed two evolutionary pathways of resistance involving the *RPA190* gene. The results of their research indicate that changes in the activity of *Phytophthora* RNA polymerase are important resistance mechanisms. Similar results were also confirmed in *P. infestans*. The diversity of the RNA polymerase I large subunit sequence of *P. infestans* plays a key role in its resistance to metalaxyl [15]. The biological degradation of metalaxyl by *Phytophthora* is another resistance mechanism. The RNA polymerase activity of sensitive strains was shown to be significantly inhibited by metalaxyl, while resistant strains showed only a slight inhibitory effect and had a certain degradation effect on metalaxyl [14].

Although crop losses caused by *P. capsici* have increased in recent years, we know very little about the molecular basis of its pathogenicity in peppers. Therefore, excavation of metalaxyl-resistance genes of *P. capsici* at the molecular level has become an important means of developing resistance to pepper blight. With the decrease in the cost of high-throughput sequencing, transcriptome sequencing has been widely used in molecular biology research and has become one of the most commonly used high-throughput sequencing technologies [16,17]. It has a number of advantages, such as a wide range of applications, good repeatability, good sensitivity, and high sequencing throughput. It can be used to discover new genes, optimize structural genes, and analyze differential expression of different transcripts, making it very convenient for differentially expressed gene (DEG) screening. The genome of *P. capsici* LT1534 was sequenced in 2012 and its size is 64 MB [18]. Transcriptome information was compared against the known genome sequence of *P. capsici*, and then the sequencing depth and expression of DEGs were analyzed to predict new genes and identify alternative splicing and gene fusion. This is of great significance for elucidating the resistance mechanism of *P. capsici* to pesticides. Parada-Rojas et al. [19] identified and characterized microsatellites in the *P. capsici* transcriptome, and then assayed a subset of 50 microsatellites in a diverse set of *P. capsici* isolates to find polymorphism. Their findings revealed that 12 microsatellites were useful to characterize the population structure of *P. capsici* and were potentially transferable to closely-related *Phytophthora* spp. Root rot caused by *P. capsici* is the most serious disease in black pepper. Researchers performed transcriptome analysis to identify candidate genes for field tolerance to black pepper root rot [20]. Sequence analysis revealed a series of proteins involved in black pepper tolerance to root rot, including signal proteins and defense enzymes such as premnaspirodiene oxidase, a phosphatase 2C-like domain protein, a mature protein of the nitrous oxide reductase family, disease resistance protein RGA3, asparaginase, β-glucosidase, a cytochrome P450 signal protein, serine/threonine protein kinase WAG1, and nucleoredoxin 1-1 enzyme. Chen et al. [21] applied RNA-Seq technology to reveal a large number of genes related to pathogenicity at three stages of mycelia (MY), zoospores (ZO), and germinating cysts with germ tubes (GC) were identified, including 98 predicted effector genes. Therefore, transcriptome sequencing can be used to study the gene functions of pathogenic bacteria and analyze the expression differences between different strains to screen genes related to target traits.

In this study, the sensitive strain SD1 and the resistant mutant strain SD1-9 were used as test materials, and their transcriptomes were sequenced and analyzed. Referring to the published *P. capsici* LT1534 genome, we performed data splicing, gene expression analysis, and function prediction. Our results will help to understand the molecular mechanism of metalaxyl resistance in *P. capsici*, provide a theoretical basis for the monitoring and treatment of metalaxyl resistance of *P. capsici*, and provide an important reference for studying the molecular mechanism of metalaxyl resistance in other *Phytophthora* species.

## 2. Results

### 2.1. Obtainment of the Resistant P. capsici Mutant Strain

The test strain SD1 was cultured on a 10% V_8_ plate containing 10 µg·mL^−1^ metalaxyl for two weeks. The resulting rapidly growing sector (Figure 1) was transferred onto a fresh 10% V_8_ plate containing 10 µg·mL^−1^ metalaxyl for normal growth. This strain was identified as the metalaxyl resistant mutant SD1-9. The results suggest that *P. capsici* was susceptible to high metalaxyl application induced metalaxyl resistance.

### 2.2. Determination of Pathogenicity

*P. capsici* strains SD1 and SD1-9 were inoculated onto green bell pepper, squash, cucumber, red pepper, and purple eggplant fruits. Disease spots appeared after 3 days (Figure 2), indicating that *P. capsici* could infect these five fruits. Moreover, the pathogenicity of the strain SD1 and the metalaxyl-resistant mutant strain SD1-9 differed on each vegetable, and obvious white colonies were produced on green bell pepper, cucumber, and purple eggplant. Table 1 shows the average diameter of lesions was 3.72–6.60 cm. The pathogen had the strongest pathogenicity on purple eggplant. Red pepper produced a small number of white colonies after being infected by the pathogen, showing a soft rot, and the parent strain SD1 had higher pathogenicity than the mutant strain. The pathogen showed the lowest pathogenicity on squash. Analysis of the data with the SPSS statistical software showed that each strain had different pathogenicity on different vegetables.

*P. capsici* strains SD1 and SD1-9 were treated with metalaxyl at concentrations of 0, 5, and 100 µg·mL^−1^, and their pathogenicity on green peppers and pepper leaves was consistent (Figure 3). It can be seen from Table 2 that the pathogenicity of strain SD1 was stronger than that of SD1-9 on green peppers and pepper leaves. With increasing metalaxyl concentration, the pathogenicity of *P. capsici* gradually decreased, indicating that metalaxyl can effectively inhibit *P. capsici*.

### 2.3. Transcriptome Sequencing Quality Analysis

To study the molecular mechanism of *P. capsici* resistance to metalaxyl, six transcriptomes were analyzed for resistance genes. SD1 (CK) and SD1-9 libraries were created in triplicate, and the transcriptome statistics of the six processed samples were analyzed by the Illumina sequencing method (as shown in Table 3). After low-quality reads were filtered out, the clean reads of the SD1 sequencing libraries were in the range of 38,634,410–73,337,800, and the clean reads of the SD1-9 sequencing libraries were in the range of 56,374,338–109,537,004. The clean reads were then further filtered to obtain high-quality reads (High Quality Clean Reads, referred to as HQ Clean Reads). The HQ Clean Reads of the SD1 and SD1-9 sequencing libraries were in the ranges of 37,789,918–72,092,776 and 55,305,346–107,700,064, respectively, accounting for more than 97% of the clean reads. The GC content was between 56.00% and 57.31%. In addition, the Q30 value was greater than 95%, indicating that RNA-Seq sequencing data was of good quality and could be used for bioinformatics analysis. However, depending on sample quality and species, ribosomal RNA may not be completely removed during transcriptome analysis. To avoid contamination from ribosomal RNA affecting subsequent analysis, the reads comparison tool bowtie2 (2.2.8) was used to compare the HQ Clean Reads to the ribosomal RNA of *P. capsici* (mismatch number: 0) to remove ribosomal RNA reads; the remaining data were used for subsequent analysis. The SD1 reads matching ribosomal sequences ranged from 2,118,438 to 5,264,802, which accounted for 5.05% to 7.30%; the unmatched reads ranged from 35,671,480 to 66,827,974, which accounted for 92.70% to 94.95%. The SD1-9 reads matching ribosome sequences ranged from 2,101,888 to 3,900,268, accounting for 3.62% to 4.13%; the unmatched reads ranged from 53,203,458 to 103,799,796, accounting for 95.87% to 96.38%.

Using the comparison software Tophat2 (2.1.1), the reads not aligned to ribosomal RNA were compared to the *P. capsici* LT1534 genome (Table 4). The numbers matched for SD1 were in the range of 23,943,117–44,456,782, with a comparison rate of 67.12%–67.26%; the numbers matched for SD1-9 were in the range of 37,397,410–73,232,460, with a comparison rate of 70.18%–70.55%. The reason for the low comparison rate may be that the V8 medium was contaminated during the sampling process or other impurities were introduced during the washing of the hyphae.

### 2.4. Analysis of Principal Components

To investigate the replication of the transcriptome samples, we performed principal component analysis (PCA), and the results are shown in Figure 4. The PCA results clearly divided the transcriptome samples into two groups: the control group (SD1-1, SD1-2, SD1-3) and the SD1-9 sample group (SD1-9-1, SD1-9-2, SD1-9-3). The three replicate samples within each group were brought together to form an independent population. According to the numerical values of the sample gene expression in the first principal component (PC1) and the second principal component (PC2), a two-dimensional coordinate map of the principal components was drawn. PC1 (84.2%) and PC2 (8.3%) revealed a change in gene expression of the six samples of 92.5% and showed good agreement between sample biological replicates.

### 2.5. Effect of Metalaxyl on Gene Expression of P. capsici

In the metalaxyl-treated mutant strain, the expression of metalaxyl-resistance genes of *P. capsici* changed significantly. In the SD1-9 resistant mutant strain, 517 differentially expressed genes (DEGs) were upregulated, while 3328 DEGs were downregulated (Figure 5).

After analysis of the difference in gene expression, FDR < 0.05 and |log2FC| > 1 were selected as screening criteria for comparison. The top 20 gene IDs in the samples that were significantly upregulated and downregulated are listed in Table 5 and Table 6, respectively. In Table 5, the gene descriptions for genes with significantly downregulated expression include a hypothetical protein, a conserved hypothetical protein, 5-methyltetrahydropteroyltriglutamate-homocysteine S-methyltransferase, a potential polyprotein, quinone oxidoreductase 2, serine/threonine-protein kinase drkB, acyl-coenzyme A oxidase, and a cyst germination-specific acidic repeat protein. In Table 6, the gene descriptions for genes with significantly upregulated expression include a hypothetical protein, Nef-associated protein 1, exportin-5, ABC transporter G family member 2, a C-factor, pol polyprotein fruit fly (*Drosophila melanogaster*) transposon, and NPP1 protein. These gene descriptions indicate the resistance of *P. capsici* to metalaxyl is a complicated process.

### 2.6. Gene Ontology (GO) Analysis

To elucidate the stress response of *P. capsici* to metalaxyl, we performed a GO functional enrichment analysis on the DEGs. GO is a classification system that describes the functions of genes and the relationships between genes. GO includes three ontologies: molecular function, cellular component, and biological process [22,23]. In the SD1-9 samples, 874 upregulated and 4536 downregulated DEGs were annotated, for a total of 5410 DEGs (Appendix A). Of these, 2464 DEGs (397 up, 2067 down) belonged to the biological process category, 1207 DEGs (229 up, 978 down) belonged to the cellular component category, and 1739 DEGs (248 up, 1491 down) belonged to the molecular function category (Figure 6 and Appendix A). In the biological process ontology, the functions of the genes were related to biological adhesion, biological regulation, cellular component organization or biogenesis, cellular processes, developmental processes, localization, locomotion, metabolic processes, multi-organism processes, multicellular organismal processes, responses to stimulus, signaling, and single-organism processes. The largest proportions of genes were in the cellular process, metabolic process, and single-organism process categories. Among them, 98 genes were significantly upregulated and 573 genes were significantly downregulated in the cellular process category; 132 genes were significantly upregulated and 613 genes were significantly downregulated in the metabolic process category; and 94 genes were significantly upregulated and 488 genes were significantly downregulated in the single-organism process category. In the cellular component ontology, the gene functions were related to the cell, cell part, macromolecular complex, membrane, membrane part, membrane-enclosed lumen, organelle, organelle part, virion, and virion part. The largest proportions of genes were in the cell and membrane categories; 42 genes were significantly upregulated and 164 genes were significantly downregulated in cell and cell part, 42 genes were significantly upregulated and 214 genes were significantly downregulated in membrane, and 41 genes were significantly upregulated and 200 genes were significantly downregulated in membrane part. In the molecular function ontology, the functions appearing most frequently were antioxidant activity, binding, catalytic activity, molecular function regulator, molecular transducer activity, nuclear acid binding transcription factor activity, signal transducer activity, structural molecule activity, and transporter activity. Among them, the main functions were binding and catalytic activity; 88 genes were significantly upregulated and 567 genes were significantly downregulated in binding, and 129 genes were significantly upregulated and 789 genes were significantly downregulated in catalytic activity. Genes are often associated with multiple different functions, such as the significantly downregulated gene XLOC_020226, which was associated with single-organism process, localization, cellular process, and transporter activity functions. GO analysis showed that genes were significantly upregulated and downregulated in the five categories single-organism process, cellular process, metabolic process, catalytic activity, and binding.

### 2.7. Kyoto Encyclopedia of Genes and Genomes (KEGG) Enrichment Analysis

The Q-value in KEGG enrichment analysis ranges from 0 to 1; the closer to zero, the more significant the enrichment. Universally regulated genes were enriched in the Caffeine metabolism (ko00232), Peroxisome (ko04146), Endocytosis (ko04144), beta-Alanine metabolism (ko00410), Tyrosine metabolism (ko00350), Inositol phosphate metabolism (ko00562), Fatty acid degradation (ko00071), Pyruvate metabolism (ko00620), Valine, leucine, and isoleucine degradation (ko00280), Regulation of mitophagy-yeast (ko04139), Ribosome biogenesis in eukaryotes (ko03008), and other metabolic pathways (Figure 7 and Appendix A).

### 2.8. Validation of RNA-Seq Data

To verify the reliability of the RNA-Seq data, we randomly selected 10 genes in the SD1-9 group for Quantitative Real-Time PCR (qRT-PCR) differential expression verification. Fold changes calculated from qRT-PCR were compared with the RNA-Seq expression analysis data, and the results are shown in Figure 8. The qRT-PCR results were in agreement with the RNA-Seq high-throughput sequencing data, indicating similar expression patterns of up- and downregulated genes in the RNA-Seq and qRT-PCR tests. XLOC_001584, XLOC_005234, XLOC_018749, fgenesh1_pg.C_scaffold_14000251 (14000251), gw1.17.333.1 all showed increased expression, while XLOC_018738, XLOC_020738, XLOC_011516, e_gw1.14.281.1, estExt_fgenesh1_kg.C_180404 (180404) all showed decreased expression (180404). However, the gene expression levels were different, which may have been caused by errors in the experimental process or differences in the quantitative instruments.

### 2.9. Expression of Candidate Gene XLOC_020226 During the Life History of P. capsici

The expression of the candidate gene XLOC_020226 in different life history stages is shown in Figure 9. The expression of XLOC_020226 gradually increased in the SD1 strain throughout the 10 stages. In the SD1-9 strain, the expression level reached a maximum at the ZO stage and basically showed a gradually increasing trend with increasing infection time in pepper leaves. We speculate that the XLOC_020226 gene may be related to the growth and pathogenesis of *P. capsici*; it may also be involved in regulating the zoospore release of *P. capsici*.

## 3. Discussion

Pathogenicity refers to the strength of the pathogenic infection of the host plant. There are differences in pathogenicity among different strains of *Phytophthora* species, which is manifested by different pathogenicities of strains in the same *Phytophthora* species to the same host plant and different pathogenicity of the same strain to different host plants. In traditional biology, the pathogenic inoculation methods for *Phytophthora* are root inoculation, stem inoculation, and leaf spraying. These methods are relatively simple and fast, but are susceptible to variability due to factors such as host plants, inoculation methods and environmental conditions, which can make their results differ [24]. Therefore, when measuring the pathogenicity of *Phytophthora* on host plants under laboratory conditions, the experimental conditions must be consistent to make the results accurate and reliable. Tian et al. [25] determined the virulence of *P. capsici* isolates on different pumpkin varieties, and reported that *P. capsici* isolates were more virulent towards jack-o-lantern pumpkins than processing pumpkins. Similarly, the pathogenicity test results of five *P. capsici* strains from different regions of Guangdong on eight hot (sweet) pepper materials indicated that the pathogenicities of different strains were significantly different, with differences in pathogenicity in the same strain in different pepper materials [26]. In this study, the results of the pathogenicity test showed that each strain had different pathogenicity on different host materials. The phenotypes of strains SD1 and SD1-9 on green peppers and pepper leaves showed that their pathogenicity gradually weakened as the metalaxyl concentration increased.

Previous genetic studies have shown that insensitivity to metalaxyl is regulated by one or two major MEX loci, while other genes are less affected [27,28,29,30,31]. Mionor sites that cause chemical insensitivity may include non-specific efflux pumps and detoxification. These functions can be performed by proteins such as the ATP-binding cassette (ABC) transporter and the cytochrome P450 protein, respectively [32,33]. However, *P. infestans* isolates that have essentially non-specific insensitivity to chemicals have not shown a corresponding increase in transcriptional abundance from genes encoding ABC transporters [29]. Transcriptome sequencing was used to analyze the gene functions and expression of the sensitive strain SD1 and metalaxyl-resistant mutant strain SD1-9. The results showed that compared with the sensitive strain SD1, the mutant strain SD1-9 treated with metalaxyl had 517 significantly upregulated genes and 3328 significantly downregulated genes. The functions of these DEGs varied. Researchers [34,35,36] have used ^3^H uridine incorporation biochemical analysis to show that metalaxyl has specific effects on the synthesis of RNA, especially ribosomal RNA (rRNA), while messenger (mRNA) and transport RNA (tRNA) synthesis is less affected. These effects are related to RNA polymerase I (RNApolI) because it transcribes rRNA. In addition, another study showed that metalaxyl exerts its activity when the RNA polymerase complex binds to DNA [34]. RNA polymerases that rely on eukaryotic DNA are multi-subunit complexes, and up to seven subunits can be shared among the three major RNA polymerases [37,38]. Other proteins such as topoisomerases and transcription factors may also affect RNA polymerase activity [39]. These factors complicate the identification of metalaxyl-targeted RNA polymerase subunits and sequence variations that lead to insensitivity.

## 4. Materials and Methods

### 4.1. Tested Strain

*P. capsici* strain SD1 was isolated from diseased plants with the typical symptoms of pepper blight. The plants were collected from a metalaxyl-free pepper field in Taian, Shandong Province. According to the metalaxyl sensitivity test, the EC_50_ value was 0.4 µg·mL^−1^, indicating that the strain was sensitive to metalaxyl. The strain has been deposited to the Fungal Laboratory of Anhui Agricultural University.

Generation of the metalaxyl-resistant mutant strains: The metalaxyl-sensitive *P. capsici* strain SD1 was used as the wild type strain. After the strain was cultured on 10% V_8_ (V_8_ juice 10 mL, H_2_O 90 mL, CaCO_3_ 0.02 g, agar 3 g) plates for 5–7 d, the mycelium discs with a diameter of 6–10 mm were transferred onto 10% V_8_ solid medium containing 10 µg·mL^−1^ metalaxyl [40], and incubated in a 25 °C incubator. After 7 d of incubation, the growth of the colonies was observed. After 10–14 d of incubation, if a rapidly growing sector (mycelial growth rate > 6 mm·d^−1^) appeared, the colonies in that sector were transferred onto 10% V_8_ solid medium containing 10 µg·mL^−1^ metalaxyl. A normal mycelial growth rate (3–6 mm·d^−1^) was considered to be a sign of acquisition of metalaxyl-resistance and such strains were chosen for further study.

### 4.2. Pathogenicity Determination

Pathogenicity on different vegetables: Green bell peppers, squash, red peppers, purple eggplants, and cucumbers that were fresh, healthy, and basically the same size were purchased from a local market in Hefei. Strains SD1 and SD1-9 were transferred to 10% V_8_ medium and cultured in the dark at 25 °C for 5–7 days. Mycelium discs with a diameter of 6–10 mm were cut out. Three small insect needles were used to puncture the surface of each vegetable, and the discs were applied with the mycelium side facing the wound. Absorbent cotton was dipped in water and placed over the wound for 24 h, and each treatment was repeated four times, with strain SD1 as a control. The samples were then cultured in a 25 °C light incubator, and lesion size was measured after 72 h.

Pathogenicity under different metalaxyl concentrations: Pepper leaves (variety Xinsujiao 5) and green peppers that were fresh, healthy and consistent in size were selected. Strains SD1 and SD1-9 were transferred to 10% V_8_ medium and cultured in the dark at 25 °C for 5–7 days. Mycelium discs with a diameter of 6–10 mm were cut out. Three small insect needles were used to puncture the surfaces of the pepper leaves and green peppers, and the discs were applied with the mycelium side facing the wound. Each wound was sprayed with 1 mL of metalaxyl medicinal solution, at concentrations of 0, 5, and 100 µg·mL^−1^. After this treatment, absorbent cotton was put on the wound and moistened for 24 h. Each treatment was repeated four times. A blank 10% V_8_ medium block was inoculated and sprayed with sterile water as a control. After inoculation, the samples were cultured in a 25 °C light incubator. Lesion size was measured after 72 h and the test was repeated twice.

### 4.3. RNA Extraction, cDNA Library Preparation and Transcriptome Sequencing

The *P. capsici* strains SD1 and SD1-9 were transferred to 10% V_8_ medium and cultured in the dark at 25 °C for 5–7 days. Mycelium discs with a diameter of 5 mm were transferred to 10% V_8_ liquid medium (V_8_ juice 10 mL, H_2_O 90 mL, CaCO_3_ 0.02 g). Ten mycelium discs per dish and 10 dishes per strain were placed in the dark at 25 °C for 3 days. After 3 days, the mycelium was filtered with gauze, rinsed with 25% ethanol, and repeatedly washed with deionized water three to four times. The excess water was squeezed out to collect mycelia. The collected mycelia were quickly frozen and ground into powder with liquid nitrogen for RNA extraction. Each strain was repeated three times. RNA extraction was performed using the Takara RNA extraction kit, following the method provided. The concentration and purity of the RNA samples were quantified using a spectrophotometer (NanoDrop ND-1000; Thermo Fisher Scientific, Waltham, MA, USA), and the RNA degradation of the six samples was assessed in 1% agarose gels. The RNA integrity was assessed using the RNA Nano 6000 Assay Kit with the Agilent Bioanalyzer 2100 system (Agilent Technologies, CA, USA). Sequencing libraries were generated using a NEBNext Ultra™ RNA Library Prep Kit for Illumina (NEB, Ipswich, MA, USA), following the manufacturer’s recommendations. The effective concentration of each library was accurately quantified using qPCR to ensure library quality, and then cDNA library sequencing was conducted with an Illumina high-throughput sequencing platform (HiSeq™ 2500) by Genedenovo Biotechnology Co., Ltd. (Guangzhou, China).

### 4.4. RNA-Seq Data Analysis

The constructed libraries were sequenced with an Illumina HiSeq™ 2500. After filtering the sequencing data to obtain clean data, the reads were compared to the *P. capsici* LT1534 genome and Cufflinks was used to splice the reads to obtain transcript data. Then, the obtained genes were statistically analyzed, and differential expression and functional enrichment analyses were performed. The edgeR software was used to analyze the differences in gene expression between groups. FDR (corrected *p*-value, indicating significance) and log_2_FC (FC is the fold change multiple) were used to screen for DEGs. The screening criteria were FDR < 0.05 and |log_2_FC| > 1. The library corresponding to the SD1 sample was used as a control.

### 4.5. Gene Annotation

GO annotation is based on the significant enrichment of GO functions to analyze DEGs and related gene modules for bioinformatics analysis. Referring to the annotation information in the NCBI non-redundant (Nr) database, the Blast2 GO software (version 3.0, https://www.blast2go.com/, BioBam Bioinformatics S.L., Valencia, Spain) was used to perform GO annotation on the core DEGs and the co-expressed gene modules, and the WEGO software was used to annotate and statistically analyze the GO functional classifications of all genes. GO covers three aspects of biology: cellular components, molecular functions, and biological processes. The KEGG database (https://www.kegg.jp/) can systematically classify and annotate the metabolic pathways of genes and can be used to study genes and their expression information at a general level. The KO-BAS software (version 2.0, KOBAS, Surrey, UK) was used to detect the enrichment of DEGs in KEGG pathways, and the biological functions of specific genes of *P. capsici* were considered and evaluated at a macro level.

### 4.6. qRT-PCR

The remaining *P. capsici* SD1 and SD1-9 samples were taken for RNA extraction and purification treatment experiments and then reverse transcribed into cDNA. Ten differentially expressed genes (including genes that were upregulated and downregulated) were randomly selected, and their relative expression levels were verified using a quantitative real-time PCR method. Primer 3.0 was used to design qRT-PCR primers online. The gene names and quantitative primers are shown in Appendix A. Quantification was performed following the operation method of the CFX96 Real-time PCR Detection System and the TB Green™ Premix Ex Taq™ II kit. Each analysis consisted of three biological replicates and three technical replicates per biological replicate. Quantitative fluorescence analysis was performed using the *β-actin* gene of *P. capsici* as an internal reference. The mRNA levels were normalized with the relative mRNA level of the *P. capsici β-actin* gene using the 2^−ΔΔCt^ method [41].

### 4.7. RNA Extraction at Various Growth Stages of P. capsici

Total RNA was extracted from *P. capsici* SD1 and SD1-9 at 10 stages, including the mycelia (MY), zoosporangia (SP), zoospores (ZO), cysts (CYST), germinating cysts (GC), and infected pepper leaves at 1.5, 3, 6, 12, and 24 h. The RNA extraction method in each period followed Zhang [42]. The cDNA was synthesized using the TaKaRa PrimeScript™ RT reagent Kit with gDNA Eraser and used as the template for qRT-PCR. The qRT-PCR primers are listed in Appendix A.

## Figures and Tables

**Figure 1 microorganisms-08-00278-f001:**
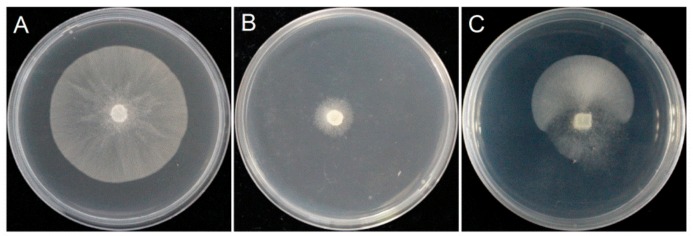
Colony picture of *P. capsici*. (**A**): Colony morphology of *P. capsici* SD1 on 10% V_8_ medium; (**B**): *P. capsici* SD1 does not produce mutants on metalaxyl-containing 10% V_8_ medium; (**C**): The *P. capsici* SD1 mutant on metalaxyl-amended 10% V_8_ medium (mutant sector).

**Figure 2 microorganisms-08-00278-f002:**
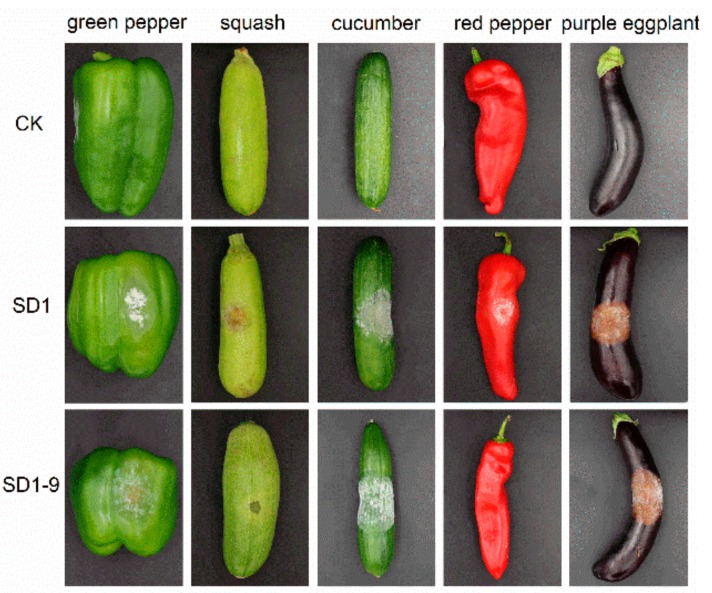
Pathogenic phenotype of *P. capsici* on different vegetables.

**Figure 3 microorganisms-08-00278-f003:**
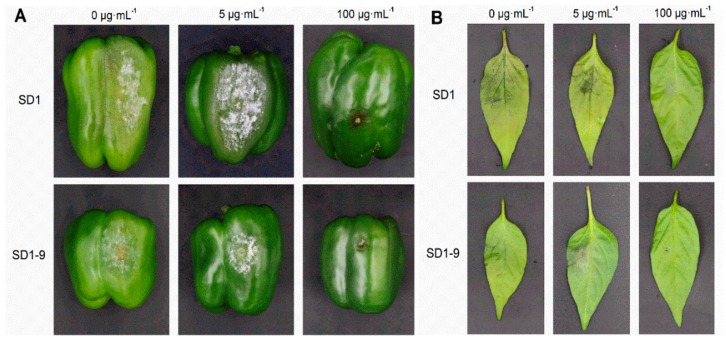
Pathogenicity of *P. capsici* under different metalaxyl concentrations. (**A**): Pathogenicity of *P. capsici* on green peppers. (**B**): Pathogenicity of *P. capsici* on pepper leaves.

**Figure 4 microorganisms-08-00278-f004:**
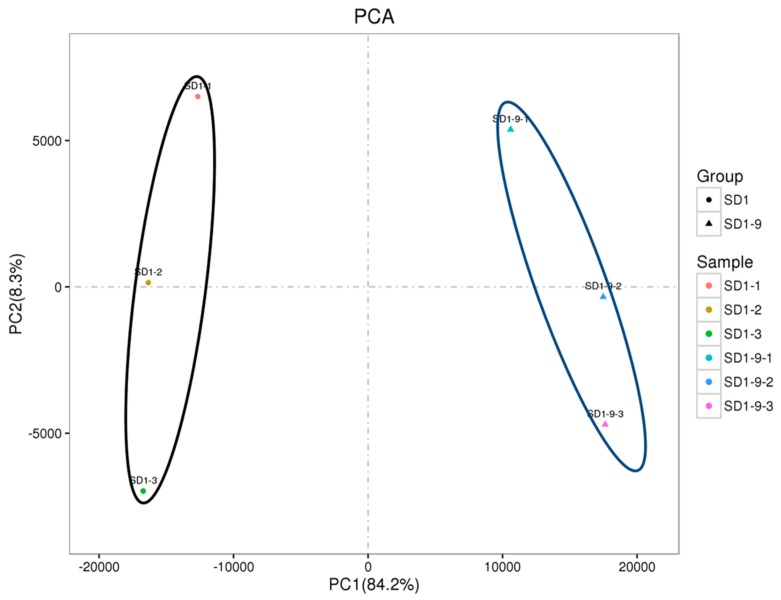
Principal component analysis of the *P. capsici* samples. The black circle represents the control 1 (SD1-1), control 2 (SD1-2), and control 3 (SD1-3) samples. The blue circle represents *P. capsici* SD1-9 samples; SD1-9-1, SD1-9-2, and SD1-9-3 are the three biological repeats.

**Figure 5 microorganisms-08-00278-f005:**
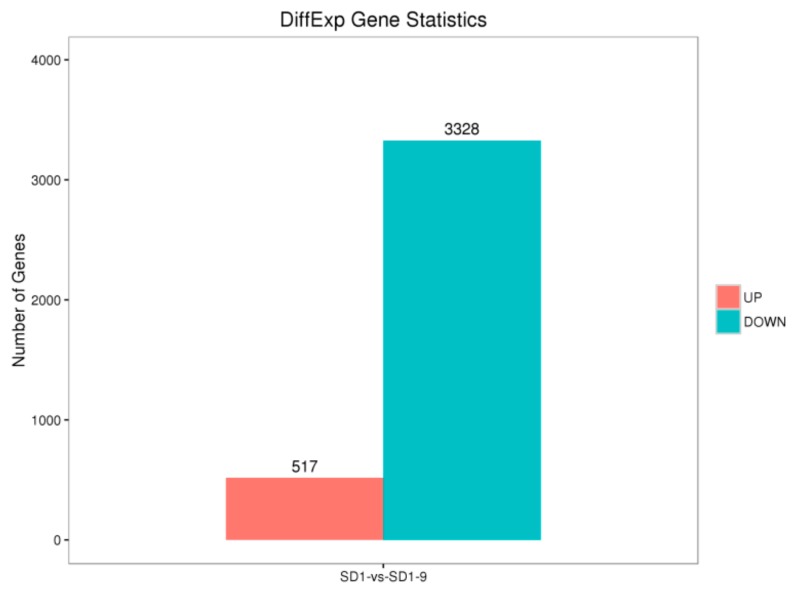
Number of differentially expressed genes of *P. capsici*.

**Figure 6 microorganisms-08-00278-f006:**
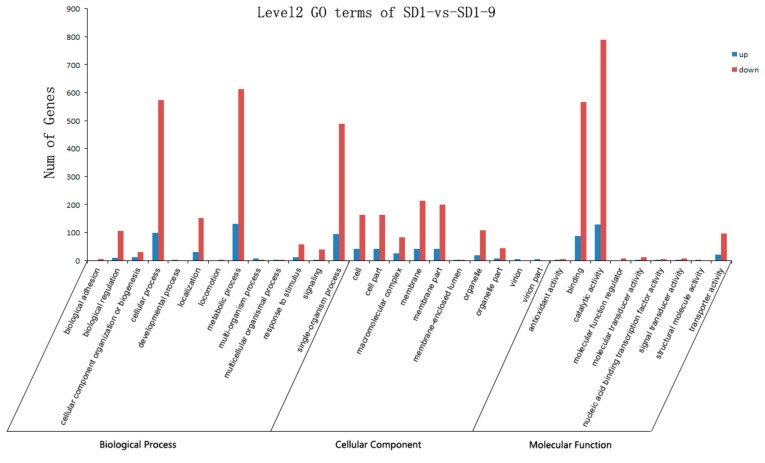
Gene Ontology (GO) enrichment analysis for differentially expressed gene (DEG).

**Figure 7 microorganisms-08-00278-f007:**
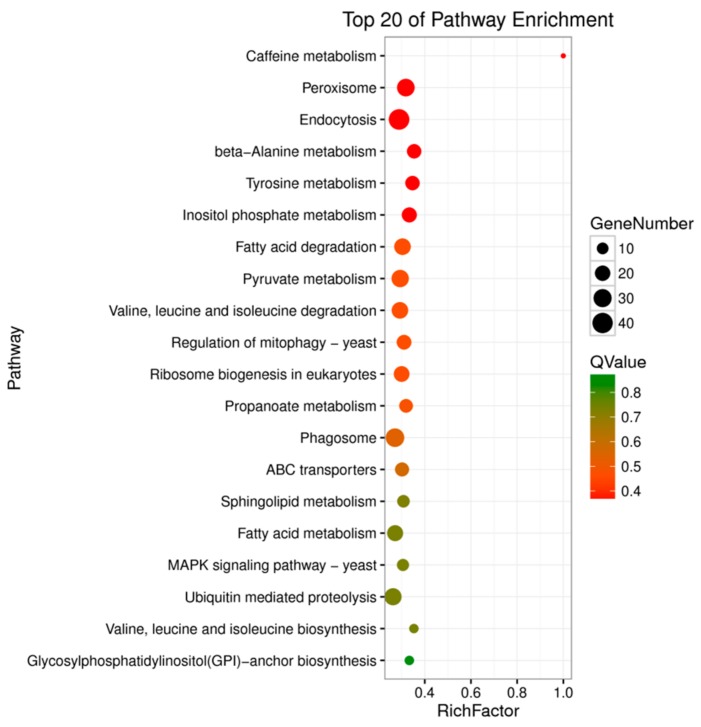
KEGG pathway enrichment analysis for DEGs of *P. capsici*.

**Figure 8 microorganisms-08-00278-f008:**
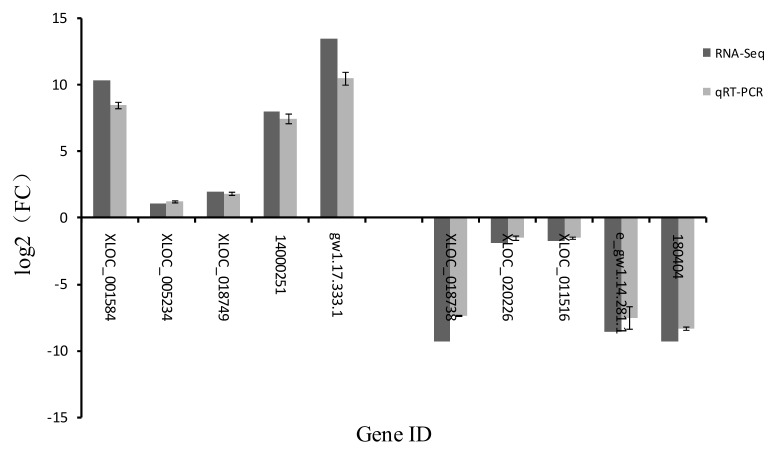
Comparison of gene expression patterns obtained using RNA-Seq and qRT-PCR. The *X*-axis shows genes validated in this study; the *Y*-axis shows the log2 ratio of expression in SD1-9 versus SD1.

**Figure 9 microorganisms-08-00278-f009:**
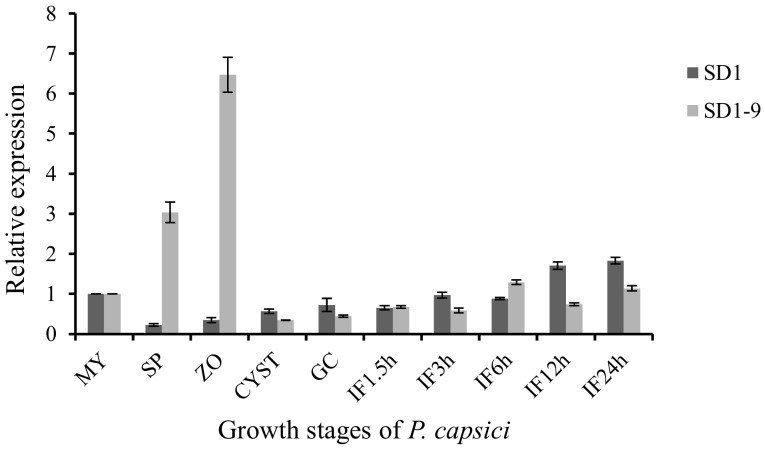
Expression of gene XLOC_020226 during the life cycle of *P. capsici*.

**Table 1 microorganisms-08-00278-t001:** Pathogenicity of the metalaxyl-sensitive and -resistant mutant strains of *P. capsici* on different vegetables.

Strain	Green Pepper (cm)	Squash (cm)	Cucumber (cm)	Red Pepper (cm)	Purple Eggplant (cm)
SD1	4.15 ± 0.07a	3.00 ± 0.21a	5.25 ± 0.81a	5.35 ± 0.07a	6.10 ± 0.26a
SD1-9	3.72 ± 0.25b	1.20 ± 0.05b	6.92 ± 0.38b	4.43 ± 0.27b	6.60 ± 0.26b

Note: In the same column, the same lowercase letter indicates that the difference is not significant, while different lowercase letters indicate that the difference is significant (*p* < 0.05).

**Table 2 microorganisms-08-00278-t002:** Pathogenicity of *P. capsici* on green peppers and pepper leaves under different metalaxyl concentrations.

Strain	Average Diameter of Lesions on Green Peppers (cm)			Average Diameter of Lesions on Pepper Leaves (cm)		
	0 µg·mL^−1^	5 µg·mL^−1^	100 µg·mL^−1^	0 µg·mL^−1^	5 µg·mL^−1^	100 µg·mL^−1^
SD1	4.13 ± 0.38a	3.93 ± 0.31a	1.33 ± 0.15a	3.27 ± 0.24a	2.57 ± 0.07a	0.93 ± 0.06a
SD1-9	3.72 ± 0.28b	2.9 ± 0.23b	1.28 ± 0.06a	2.50 ± 0.05b	1.90 ± 0.05b	0.57 ± 0.03b

Note: In the same column, the same lowercase letter indicates that the difference is not significant, while different lowercase letters indicate that the difference is significant (*p* < 0.05).

**Table 3 microorganisms-08-00278-t003:** Summary of the quality of transcriptomic sequencing data of *P. capsici*.

Sample	Clean Reads Num	HQ Clean Reads Num (%)	Read Length	Adapter (%)	Low Quality (%)	Q30(%)	GC (%)	Mapped Reads	Unmapped Reads
SD1-1	47256016	46366396 (98.12%)	150 + 150	291356 (0.62%)	597932 (1.27%)	95.61%	56.62%	2339460 (5.05%)	44026936 (94.95%)
SD1-2	73337800	72092776 (98.3%)	150 + 150	423868 (0.58%)	820680 (1.12%)	96.03%	56.00%	5264802 (7.30%)	66827974 (92.70%)
SD1-3	38634410	37789918 (97.81%)	150 + 150	331822 (0.86%)	512404 (1.33%)	95.98%	56.35%	2118438 (5.61%)	35671480 (93.39%)
SD1-9-1	56374338	55305346 (98.1%)	150 + 150	431358 (0.77%)	637298 (1.13%)	96.08%	57.31%	2101888 (3.80%)	53203458 (96.20%)
SD1-9-2	109537004	107700064 (98.32%)	150 + 150	769916 (0.7%)	1066732 (0.97%)	96.36%	57.30%	3900268 (3.62%)	103799796 (96.38%)
SD1-9-3	59170962	58097958 (98.19%)	150 + 150	410054 (0.69%)	662508 (1.12%)	95.98%	57.10%	2400164 (4.13%)	55697794 (95.87%)

Note: SD1-1 = control 1, SD1-2 = control 2, and SD1-3 = control 3; SD1-9-1, SD1-9-2, and SD1-9-3 are three biological repeats of the SD1-9 group. Clean Reads: the number of reads after initial filtering; HQ Clean Reads: the number of reads obtained by further filtering of the clean reads. Mapped Reads: the number of reads mapped to ribosomal sequences; Unmapped Reads: the number of reads not mapped to ribosomal sequences.

**Table 4 microorganisms-08-00278-t004:** Read numbers after alignment with the reference genome.

Sample	Total Reads	Mapped Reads	Mapping Rate (%)
SD1-1	44026936	29610887	67.26%
SD1-2	66827974	44456782	66.52%
SD1-3	35671480	23943117	67.12%
SD1-9-1	53203458	37379410	70.26%
SD1-9-2	103799796	73232460	70.55%
SD1-9-3	55697794	39085928	70.18%

Note: SD1-1 = control 1, SD1-2 = control 2, and SD1-3 = control 3; SD1-9-1, SD1-9-2, and SD1-9-3 are three biological repeats of the SD1-9 group. Total Reads: the number of reads after excluding ribosomal RNA; Mapped Reads: the number of reads uniquely mapped to the reference genome; Mapping rate (%): comparison rate.

**Table 5 microorganisms-08-00278-t005:** Top 20 downregulated genes of *P. capsici*.

Gene id	Log2(FC)	Description
XLOC_005973	−13.1721409831293	hypothetical protein L915_18002
estExt_fgenesh1_pm.C_190011	−12.8848064878385	hypothetical protein PHYSODRAFT_549598
fgenesh1_pg.C_scaffold_14000098	−12.8211096047814	conserved hypothetical protein
XLOC_012642	−12.093417564388	conserved hypothetical protein
fgenesh1_pg.C_scaffold_4000028	−11.7827252555489	hypothetical protein L915_21744
fgenesh1_pg.C_scaffold_17000301	−11.6423525991843	-
gw1.6.121.1	−11.4935219166903	5-methyltetrahydropteroyltriglutamate-homocysteine S-methyltransferase
fgenesh1_kg.C_scaffold_19000329	−11.3275526440812	hypothetical protein F443_12900
e_gw1.9.155.1	−10.8428741027374	potential polyprotein
XLOC_003896	−10.6882503091332	-
e_gw1.2377.1.1	−10.6257088430645	conserved hypothetical protein
XLOC_015528	−10.2992080183873	hypothetical protein AM587_10012859
gw1.18.271.1	−10.0361736125535	Quinone oxidoreductase 2
e_gw1.22.342.1	−9.80305478462398	serine/threonine-protein kinase drkB
gw1.9.125.1	−9.74259014336008	potential polyprotein
gw1.15.545.1	−9.66770293172909	Acyl-coenzyme A oxidase
gw1.343.7.1	−9.4374053123073	Cyst germination specific acidic repeat protein
XLOC_000662	−9.31439442220196	conserved hypothetical protein
XLOC_018738	−9.30682120249715	hypothetical protein PPTG_10992
fgenesh1_pg.C_scaffold_4000136	−9.30682120249715	-

**Table 6 microorganisms-08-00278-t006:** Top 20 upregulated genes of *P. capsici*.

Gene id	Log2(FC)	Description
gw1.17.333.1	13.4334985082445	hypothetical protein L917_16768
fgenesh1_pm.C_scaffold_22000006	11.6721306437699	hypothetical protein F442_02861
XLOC_016746	11.1984450414524	hypothetical protein PPTG_05046
XLOC_003966	10.4512111118323	hypothetical protein F443_16013
XLOC_001584	10.3553510964248	hypothetical protein AM588_10000818
fgenesh1_pg.C_scaffold_17000275	10.1632303488683	Nef-associated protein 1
fgenesh1_pm.C_scaffold_21000228	9.85070776021539	Exportin-5
e_gw1.84.43.1	9.56877046721857	hypothetical protein L917_10475
fgenesh1_pg.C_scaffold_21000144	9.25266543245025	ABC transporter G family member 2
fgenesh1_pg.C_scaffold_1000277	9.04074634234331	hypothetical protein PHYSODRAFT_508963
estExt_Genewise1Plus.C_4230002	9.01308999944045	C-factor
gw1.88.26.1	8.98489310760979	hypothetical protein F442_02795
fgenesh1_pg.C_scaffold_18000238	8.82442843541654	-
XLOC_017752	8.77332239445924	hypothetical protein F442_04037
fgenesh1_pg.C_scaffold_1912000004	8.67948009950545	-
fgenesh1_kg.C_scaffold_3000631	8.49884920653172	-
fgenesh1_pg.C_scaffold_91000005	8.4093909361377	pol polyprotein fruit fly (Drosophila melanogaster) transposon
fgenesh1_pg.C_scaffold_23000097	8.33687843635333	NPP1 protein
fgenesh1_kg.C_scaffold_8525000005	8.33687843635333	hypothetical protein PHYSODRAFT_354574
fgenesh1_pg.C_scaffold_19000282	8.09451759878429	-

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
