# Peer review of "Differential Potential of Phytophthora capsici Resistance Mechanisms to the Fungicide Metalaxyl in Peppers"

_microorganisms, 2020, doi:10.3390/microorganisms8020278_

Round 1
Reviewer 1 Report
Crop losses caused by Phytophthora capsici have increased in recent years but we know very little about the molecular basis of its pathogenicity in papers. Authors of this article elucidate not only pathogenicity, resistant mutant strains of aforementioned pathogen but also performed transcriptome sequencing quality analysis. To better understand the stress response of P. capsici to metalaxyl (describe the functions of genes and the relationships between genes) they run GO functional enrichment analysis on DEGs. To sum up I fully recommend to publish this article in Microorganisms journal.
I congratulate the authors well written paper on pathogenicity of P. capsici, which refers to the strength of pathogenic infection of the host plant. Usually in pathogenicity tests the spores, mycelium are added either to the soil (vicinity of roots of living plant) or are inoculated directly in living plant tissues (stem, leaves etc.). In this experiment fruits were purchased from local market and mycelium discs were placed there. This simple and fast method however is sensitive to changing external conditions (we don’t know market conditions), which can make the results differ. Nevertheless, authors kept them in monitored, controlled, laboratory conditions. Even if experimental conditions were consistent and results accurate and reliable still we cannot be sure about transferring results achieved on detachable fruits on living plants (where are active resistance processes going on)? From this point of view additional tests in the future are needed to confirm this.
Line 58 Phytophthora put in Italic
Check Latin names through the text including references e.g. look at lines 489, 490
Line 547 start Phytophthora with big letter
Line 255 Disruptions in Fig. 6 is not readable at all (should be changed)
Line 279 Descriptions are overlapping with bar graphs
Line 530 add extra space after sp. and put in italics medicaginis (also in line 533)
Line 497 in big letter Xanthonomas and in italics (also twice oryzae)
Author Response
February 15, 2020
Dear reviewers:
Thank you heartily for your diligent work and good suggestions for the publication of our manuscript! We have revised the manuscript according to your suggestions, and the detailed corrections are listed below point by point:
Even if experimental conditions were consistent and results accurate and reliable still we cannot be sure about transferring results achieved on detachable fruits on living plants (where are active resistance processes going on)? From this point of view additional tests in the future are needed to confirm this.
√ You are all right. Thank you heartily for your valuable suggestion. We will do additional tests to confirm that in the future.
Line 58 Phytophthora put in Italic
√ Line 58 has been used italic for the word "Phytophthora". In line 59 of the revised manuscript.
Check Latin names through the text including references e.g. look at lines 489, 490
√ lines 489, 490 have been modified to:Comparative transcriptome analysis reveals the signal proteins and defence genes conferring foot rot (Phytophthora capsici sp. nov.) resistance in black pepper (Piper nigrum L.). In lines 493-494 of the revised manuscript.
Line 547 start Phytophthora with big letter
√ Line 547 has been used big letter for the word "Phytophthora". In line 552 of the revised manuscript.
Line 255 Disruptions in Fig. 6 is not readable at all (should be changed)
√ Figure 6 has been changed. In line 256 of the revised manuscript.
Line 279 Descriptions are overlapping with bar graphs
√ Line 279 has been modified to: Comparison of gene expression patterns obtained using RNA-Seq and qRT-PCR. The X-axis shows genes validated in this study; the Y-axis shows the log2 ratio of expression in SD1-9 versus SD1. In line 281 of the revised manuscript.
Line 530 add extra space after sp. and put in italics medicaginis (also in line 533)
√ Space have been added after sp. in line 530 and the word medicaginis has been changed to italics,included line 533. In lines 535 and 539 of the revised manuscript.
Line 497 in big letter Xanthonomas and in italics (also twice oryzae)
√ xanthonomas has been modified to Xanthonomas and uses italics.
oryzae has been modified to italics. In line 501 of the revised manuscript.
It is hoped that respected reviewers will kindly point out our errors.
With best regards.
First author: Weiyan Wang
Corresponding Author: Zhimou Gao
Reviewer 2 Report
Despite the numerous studies that have been conducted on host resistance different Phytophthora sp., only limited progress has been achieved in durably controlling the disease, in part due to the fast evolution and adaptive capacity of this pathogen. Some Oomycetes show resistance to the fungicide metalaxyl and Phytophthora capsici mutants are one of them. Transcriptome analysis of the pathogen-mutants exhibiting resistance to a fungicide is always very informative at the molecular point of view. The paper brings an answer which group of genes could be involved in resistance mechanisms in P. capsici against the fungicide.
This study suggests that metalaxyl is involved in P. capsici’ growth repression and led to important transcriptome changes by both up- and down-regulating of different gene expression in P. capsici. Among DEGs there are 5 categories of genes distinguished, associated with: single-organism process, localization, cellular process, and transporter activity functions. An interesting candidate gene (XLOC_020226) is probably involved in the regulation of the zoospore release of P. capsici. Such information is fundamental for disruption of the pathogenicity process.
The list of genes potentially involved in fungicide-resistance is worth to complete and certainly will provide an important reference for studying the molecular mechanism of metalaxyl resistance in other Phytophthora species. In a present research, it is interesting that more genes were down-regulated than up-regulated but Authors interestingly combine this fact with many factors involved (among them DEGs) in the synthesis of RNA polymerase I subunits, an enzyme well studied in cancer therapy in eukaryotes.
Summing-up, with the increasing public awareness and reluctance to use chemical pesticides, more environmentally friendly approaches are gaining in popularity and research conducted in this paper fulfills this need. I recommend publishing the article after minor corrections.
About the content:
The article is original, has good technical quality and large general interest.
The title of the article is a little bit too descriptive. It could be simplified e.g. “Differential potential of Phytophthora capsici resistance mechanisms to the fungicide metalaxyl in peppers”. Last sentence in Abstract (line 35-39) contains repetitions and needs revision. Keywords clearly reflect paper's content.
Introduction presents the problem clearly.
Results are fully presented, the mutants’ strains obtainment was explained and the pathogenicity of the P. capsici analyzed.
Discussion is short but justified.
Experimental methods are adequate and well described. The transcriptome sequencing of the SD1 and SD1-9 libraries was properly checked for ribosomal RNA contamination. All results were confirmed by adequate PCA graph, showing two separate groups of transcriptomes in tested specimens.
References are complete and adequate.
About Presentation:
Length is commensurate with the paper's content.
Quality of figures and tables is adequate except the figure 6, where the description of the OX axis is unreadable.
The English language is adequate.
About Scientific evaluation:
The general scientific approach is properly stated and well explained.
However, in my opinion some data is missing in methods - it has not been mentioned how many RNA extractions were done per strain - only one?
Some spelling errors occurred:
Lines 489-490 – the Latin names of the species should be written in italic. Please check the italic font and capital letters for all Latin names.
Line 547 – should be “Phytophthora …”
Lines 540, 545-546 – revise the spelling of the journals’ names, i.e. “Cells” or “Annual Review of Pharmacology and Toxicology”
Author Response
February 15, 2020
Dear reviewers:
Thank you heartily for your diligent work and good suggestions for the publication of our manuscript! We have revised the manuscript according to your suggestions, and the detailed corrections are listed below point by point:
The title of the article is a little bit too descriptive. It could be simplified e.g. “Differential potential of Phytophthora capsici resistance mechanisms to the fungicide metalaxyl in peppers”. Last sentence in Abstract (line 35-39) contains repetitions and needs revision. Keywords clearly reflect paper's content.
√ Based on your suggestion, the article title was modified to:Differential potential of Phytophthora capsici resistance mechanisms to the fungicide metalaxyl in peppers.
Quality of figures and tables is adequate except the figure 6, where the description of the OX axis is unreadable.
√ Figure 6 has been replaced. In line 256 of the revised manuscript.
However, in my opinion some data is missing in methods - it has not been mentioned how many RNA extractions were done per strain - only one?
√ RNA extraction was performed in triplicate for each strain.In lines 374-375 of the revised manuscript.
Lines 489-490 – the Latin names of the species should be written in italic. Please check the italic font and capital letters for all Latin names.
√ lines 489, 490 have been modified to: Comparative transcriptome analysis reveals the signal proteins and defence genes conferring foot rot (Phytophthora capsici sp. nov.) resistance in black pepper (Piper nigrum L.). In lines 493-494 of the revised manuscript.
Line 547 – should be “Phytophthora …”
√ phytophthora in Line 547 has been modified to Phytophthora. In line 552 of the revised manuscript.
Lines 540, 545-546 – revise the spelling of the journals’ names, i.e. “Cells” or “Annual Review of Pharmacology and Toxicology”
√ Line 540 journals’ names have been modified to: Cell. In line 545 of the revised manuscript.
The journals’ names of Lines 545-546 have been modified to:" Annual Review of Pharmacology and Toxicology. In lines 550-551 of the revised manuscript.
It is hoped that respected reviewers will kindly point out our errors.
With best regards.
First author: Weiyan Wang
Corresponding Author: Zhimou Gao